# Early mortality after diagnosis of cancer of the head and neck – A population-based nationwide study

**Charbél Talani[1,2], Antti Mäkitie[3,4,5], Martin Beran[6], Erik Holmberg[7], Göran Laurell[8], Lovisa Farnebo[1,2]***

**1** Division of Speech language pathology, Audiology and Otorhinolaryngology, Department of Clinical and Experimental medicine, Faculty of Health Sciences, Linköping University, Linköping, Sweden, **2** Department of Otorhinolaryngology in Linköping, Anaesthetics, Operations and Specialty Surgery Center, Region Östergötland, Linköping, Sweden, **3** Department of Otorhinolaryngology–Head and Neck Surgery, Helsinki University Hospital, and University of Helsinki, Helsinki, Finland, **4** Division of Ear, Nose and Throat Diseases, Department of Clinical Sciences, Intervention and Technology, Karolinska Institutet and Karolinska Hospital, Stockholm, Sweden, **5** Research Programme in Systems Oncology, Faculty of Medicine, University of Helsinki, Helsinki, Finland, **6** Department of ENT and Maxillofacial Surgery, NAL Medical Center Hospital, Trollhattan, Sweden, **7** Department of Oncology, Institute of Clinical Sciences, Sahlgrenska Academy at University of Gothenburg, Gothenburg, Sweden, **8** Department of Clinical Sciences, ENT, Uppsala University, Uppsala, Sweden

* lovisa.farnebo@regionostergotland.se

**Data Availability Statement:** All relevant data are within the manuscript. Data cannot be shared publicly. According to the approval of this study by the Swedish Ethical Review Authority, we are not

## Abstract

### Background

Cancers of the head and neck have a high mortality rate, and roughly 10% of the patients die within six months of diagnosis. To our knowledge little has been written about this group.

We wished to identify risk factors for early death, to predict and monitor patients at risk better and, if possible, avoid unjustified major treatment.

### Methods and findings

This population-based nationwide study from the Swedish Head and Neck Cancer Register (SweHNCR) included data from 2008–2015 and 9733 patients at potential risk of early death.

A total of 925 (9.5%) patients died within six months. For every year older the patients became, the risk of early death increased by 2.3% (p<0.001). The relative risk of death was 3.37 times higher (237%) for patients with WHO score 1 compared with WHO score 0. A primary tumour in the hypopharynx correlated with a 24% increased risk over the oral cavity (p<0.024). Patients with stage IV disease had a 3.7 times greater risk of early death than those with stage I (p<0.001). As expected, a 12 times increased risk of early death was noted in the palliative treatment group, compared to the curative group.

Limitations to this study were that the actual cause of death was not recorded in the SweHNCR, and that socioeconomic factors, alcohol consumption, smoking habits, and HPV status, were not reported in SweHNCR until 2015. However, the fact that this is a

allowed to make data available in any form other than aggregated data. Data are available from Regionalt cancercentrum väst, Sahlgrenska University Hospital 413 45 Gothenburg, Sweden. Ethics Committee approval (Gothenburg, number 299-14, T230-17) for researchers who meet the criteria for access to confidential data.

**Funding:** The authors received no specific funding for this work.

**Competing interests:** The authors have declared that no competing interests exist.

population-based nationwide study including 9733 patients compensates for some of these limitations.

## Conclusions

Identification of patients at increased risk of early death shows that older patients with advanced disease, increased WHO score, primary tumour in the hypopharynx, and those given palliative treatment, are more likely than the others to die from head and neck cancer within six months of diagnosis.

## Introduction

Sweden has a population of about 10 million inhabitants, and in 2016 the overall incidence of cancer was 64 000 in a population of just over 60 000. About 1400 new cases of cancer of the head and neck are reported annually, and according to the Swedish Cancer Society, this corresponds to 2.3% of all cancers [1]. Cancer of the head and neck is the sixth most common type worldwide, and most cases (60%) are locally advanced at the time of diagnosis (stage III or IV) [2]. Many of the tumours are aggressive, and the median survival without treatment is reported to be three to five months [3–6]. Late stages of the disease are linked to poor overall survival, but even patients with early cancers (for example of the mobile tongue) can have dismal prognoses [7, 8]. The management of metastatic and recurrent cancers of the head and neck is challenging, and typically involves combined treatments.

Both surgical and oncological treatments usually have acute side effects, but may also result in long-term consequences, and even death. Apart from our own recent review [9], the issue of early death (within six months of diagnosis) among patients with cancer of the head and neck remains poorly explored. Patients with improved survival are often HPV positive, of working age, and have only a few coexisting conditions, and for these patients reductions in treatment have been discussed [10, 11]. However, only a few authors have focussed on patients with the poorest prognosis–the older patients with coexisting conditions and high WHO scores [12, 13]. Treatment-induced side effects [14] [15] can be bearable as long as substantial duration of survival may be expected, but if it were possible to identify patients for whom it was short, resources could be more effectively spent on palliative, or at best supportive, care.

The primary aim of this study was to find out which patients were at risk of early death during the first six months after diagnosis in a large, nationwide, register-based group of patients with cancers of the head and neck. Secondly, we wanted to know whether there were any common denominators that could help identify those patients who were at risk of early death. We also aimed to increase awareness about this subgroup of patients and achieve our ultimate goal of improving the management of cancer of the head and neck.

Many studies have dealt with survival but, as far as we know, no population-based studies have focused on the patients with cancer of the head and neck who die within six months of diagnosis.

## Patients, material, and methods

We obtained our data from the Swedish Head and Neck Cancer Register (SweHNCR) (Ethics Committee approval; Gothenburg, number 299–14, T230-17), which is funded by the Swedish government, and covers 98.5% of all Swedish patients with cancers of the head and neck, when

cross-referenced with Sweden's National Board of Health and Welfare [16]. The total number of consecutive affected Swedish patients during the period 2008–2015 in the SweHNCR with one-year follow-up was 9733 (Fig 1, Table 1).

Nine sites of tumours were included: lip (C00.0–2, C00.6, C00.8, C00.9), oral cavity (C00.3, C00.4, C02, C03, C04, C05, C06), oropharynx (C01.9, C05.1, C05.2, C05.8, C05.9, C09, C10), nasopharynx (C11), hypopharynx (C12, C13), larynx (C10.1, C32), nose (C30.0) and nasal sinuses (C31), salivary glands (C07, C08), and head and neck cancer of unknown primary (C77.0), Table 1. Malignant tumours located in the thyroid, the parathyroid glands, or the oesophagus were not included. Data that are reported to SweHNCR include: incidence,

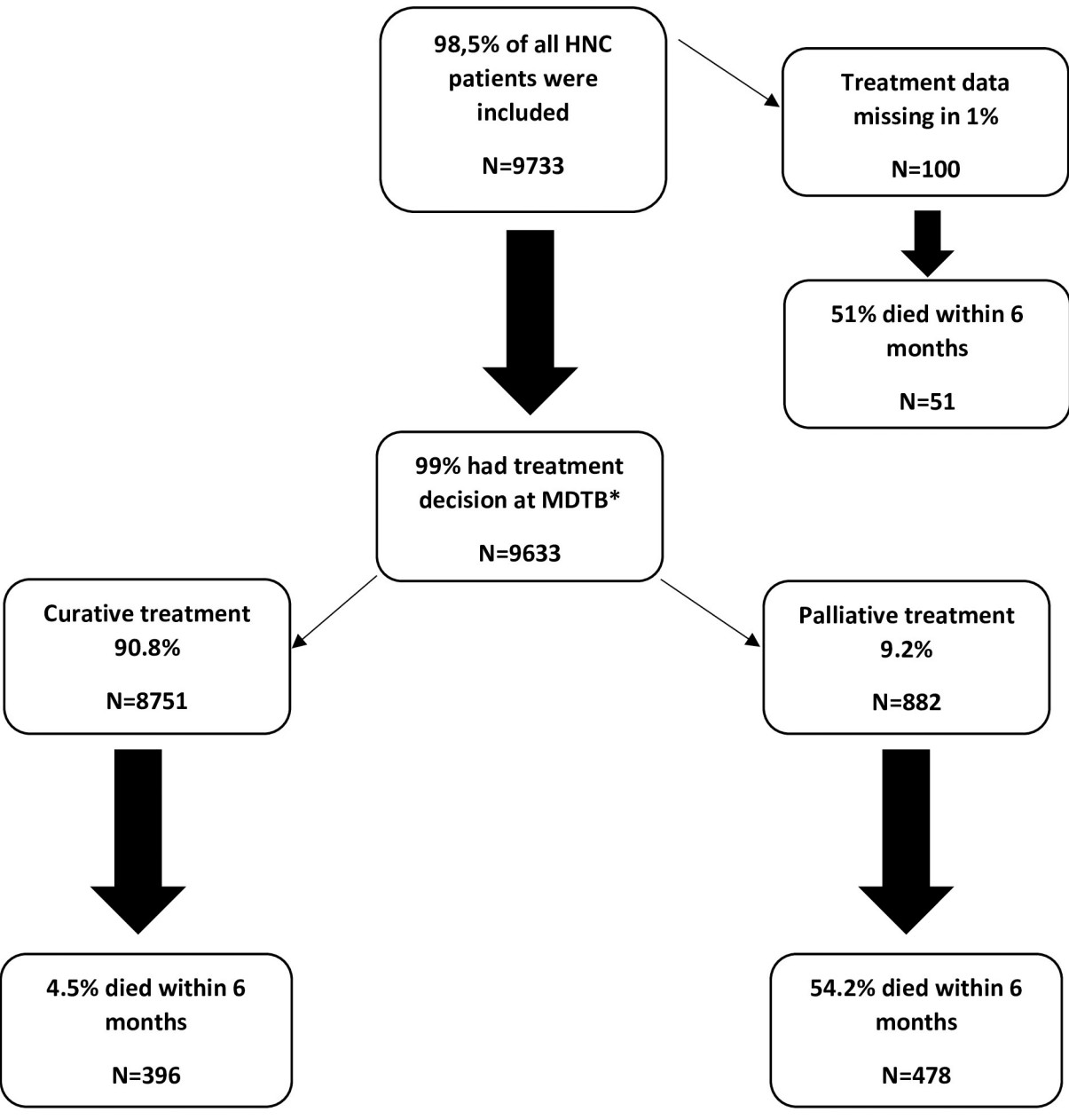

**Fig 1. Flow chart for treatment and early death for included patients.**

**Table 1. Prevalence.**

| | Prevalence n (%) Total number = 9733 |
|---|---|
| **Diagnosis** | |
| Lip | 863 (8.9) |
| Oral cavity | 2714 (27.9) |
| Oropharynx | 2524 (25.9) |
| Nasopharynx | 234 (2.4) |
| Hypopharynx | 455 (4.7) |
| Larynx | 1300 (13.4) |
| Nose/sinuses | 477 (4.9) |
| Salivary glands | 795 (8.2) |
| CUP* | 355 (3.6) |
| Incomplete diagnosis | 16 (0.2) |
| **Sex** | |
| Male | 6244 (64.2) |
| Female | 3489 (35.8) |
| **Age** continuous (mean (sd)) | 66.8 (13.3) |
| **Age groups** | |
| 0–39 | 295 (3.0) |
| 40–49 | 644 (6.6) |
| 50–59 | 1751 (18.0) |
| 60–69 | 3165 (32.5) |
| 70–79 | 2217 (22.8) |
| 80- | 1661 (17.1) |
| **Stage** | |
| I | 2564 (26.3) |
| II | 1638 (16.8) |
| III | 1265 (13.0) |
| IV | 3711 (38.1) |
| Missing | 200 (2.1) |
| (CUP lacks stage) | 355 (3.6) |
| **WHO score** | |
| 0 | 6223 (63.9) |
| 1 | 1279 (13.1) |
| 2 | 608 (6.2) |
| 3 | 344 (3.5) |
| 4 | 133 (1.4) |
| Missing | 1146 (11.8) |
| **Treatment intent** | |
| Curative | 8751 (89.9) |
| Palliative | 882 (9.0) |
| Missing | 100 (1.0) |
| **Given treatment** | |
| Surgery | 2370 (24.4) |
| Surgery+postoperative RT** | 1295 (13.3) |
| Preoperative RT+ Surgery | 306 (3.1) |
| Surgery +RT | 356 (3.7) |
| RT | 2625 (27.0) |
| CRT*** | 1169 (12.0) |

(*Continued*)

**Table 1.** (Continued)

|  | Prevalence n (%) Total number = 9733 |
| --- | --- |
| Other | 818 (8.4) |
| No treatment | 245 (2.5) |
| Missing | 549 (5.6) |

*CUP = Cancer of unknown primary

**RT = Radiotherapy

***CRT = Chemoradiotherapy

diagnosis, time to treatment, sex, age, survival, WHO score, TNM classification, stage, follow up, and recurrence. The Eastern Cooperative Oncology Group score, also called the WHO score runs from 0 to 5, with 0 denoting perfect health and 5 death [17]. The WHO score was rated 0–4, Table 2, indicating the physical performance of the patient. An overall increase in incidence of 22% was noted from 2008 (n = 1211) to 2015 (n = 1473) for head and neck cancer in Sweden, which corresponds to a yearly increase of 2.9% [16].

Treatment of these cancers in Sweden is centralised to the university hospitals, but a few second-level hospitals are able to give oncological treatment, and National Healthcare and Social Security systems are offered equally to all inhabitants. All patients were examined by either a head and neck surgeon or an oncologist after completion of treatment to evaluate its efficacy.

## Statistical methods

The primary outcome in this study was early death. The relative risk of death within 6 months of diagnosis between different groups was estimated using univariable and multivariable Poisson regression [18]. Robust variance estimates were used with Poisson models to obtain valid confidence intervals. No censuring occurred within the first 12 months. Observed survival was calculated using the Kaplan-Meier method [19]. Survival time was calculated from date of diagnosis to date of death if death occurred before 24 months of diagnosis, censored 24 months of diagnosis or earlier if end of follow-up. The patients were diagnosed in the period 2008 to June 2015 and had a follow-up until June 2016. In the survival analyses, showing probability of survival for patients with HNC based on treatment intent, tumour site, and treatment modality (Figs 2–4), patients diagnosed after June 2014 were censored at end of follow-up before reaching a 24 months follow-up if they still lived. To perform tests of differences in age, stage and WHO score between patients dead within 0–6 months, and 7–12 months compared to those alive after 12 months (Table 3), the two-sample t-test and a nonparametric test for

**Table 2.** WHO score.

| Grade | Explanation of activity |
| --- | --- |
| 0 | Fully active, able to carry on all pre-disease performance without restriction |
| 1 | Restricted in physically strenuous activity but ambulatory and able to carry out work of a light or sedentary nature, e.g., light house work, office work |
| 2 | Ambulatory and capable of all selfcare but unable to carry out any work activities. Up and about more than 50% of waking hours |
| 3 | Capable of only limited selfcare, confined to bed or chair more than 50% of waking hours |
| 4 | Completely disabled. Cannot carry on any selfcare. Totally confined to bed or chair |
| 5 | Dead |

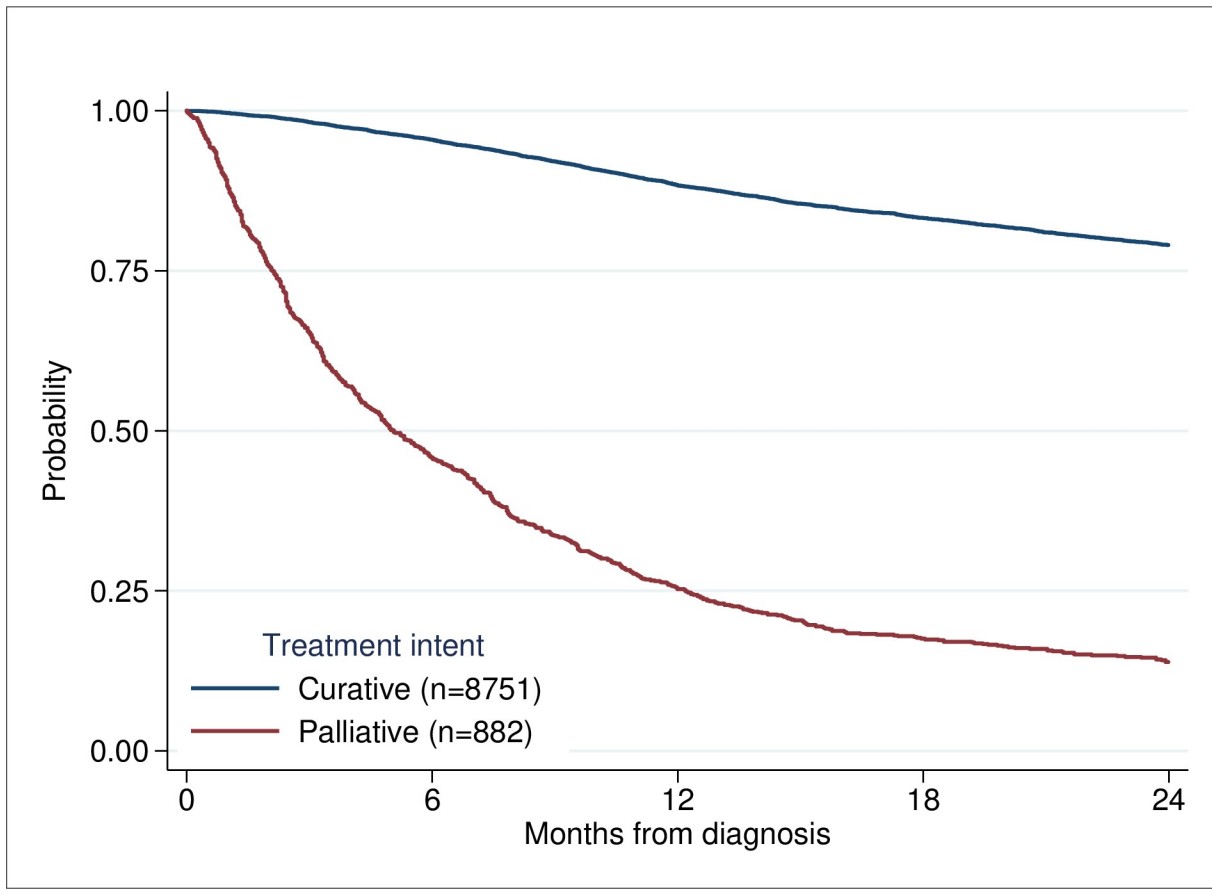

**Fig 2. Probability of survival for patients with HNC based on treatment intent.**

trend across ordered groups developed by Cuzick [20] were used. Exact binominal confidence intervals were estimated for proportions [21]. A p-value of < 0.05 was considered to be statistically significant. All statistical analyses were performed using Stata 15.1 [22].

## Results

The risk for death within six months has been divided into patient-, tumour and treatment-related factors. Of the whole group of 9733 patients, 925 died within six months of diagnosis (9.5%). Among the 9633 patients with a treatment decision at a multidisciplinary tumour board meeting, 9.1% died within six months, leaving a high rate of early death among patients for whom details of treatment were missing (n = 51, 51%) (Fig 1).

### Patient-related factors

**Sex:** 6244 patients were male and 3489 female, Table 1. There was no significant survival difference between sexes where 589 men (9.4%) and 336 women (9.6%), died within six months of diagnosis (p = 0.75), Table 4.

**Age:** the mean (range) age at diagnosis was 67 (8–102) years, and the risk to die within six months after diagnosis increased with older age, Table 3, and with 2.3% for every year older the patient became (p<0.001), Tables 5 and 6.

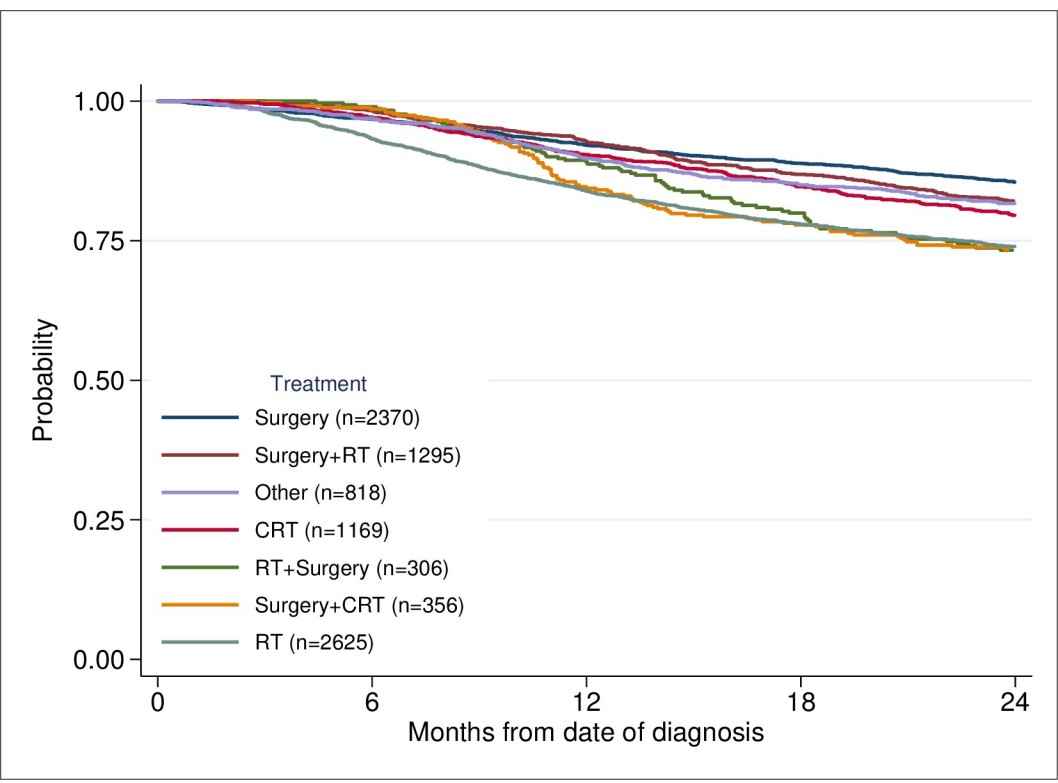

**Fig 4. Probability of survival based on treatment modality.**

**WHO score:** only 147 patients (2.4%) with a WHO score of 0 died within 6 months, compared with 166 of patients (13%) with a WHO score of 1. Of 133 patients with a WHO score of

**Table 3. WHO/Stage and survival during the first year after diagnosis.**

| | Patients dying within: | | | | Alive 12 months after diagnosis |
|---|---|---|---|---|---|
| | **0–6 months** | **p-value*** | **7–12 months** | **p-value*** | |
| **Age at diagnosis**, years; mean (SD) | 75.7 (11.1) | <0.001 | 71.8 (12.2) | <0.001 | 65.3 (13.1) |
| **WHO** | | <0.001 | | <0.001 | |
| 0 | 147 (15.9) | | 333 (41.0) | | 5743 (71.8) |
| 1 | 166 (18.0) | | 161 (19.8) | | 952 (11.9) |
| 2 | 158 (17.1) | | 130 (16.0) | | 320 (4.0) |
| 3 | 167 (18.0) | | 63 (7.8) | | 114 (1.4) |
| 4 | 107 (11.6) | | 15 (1.8) | | 11 (0.1) |
| Unknown | 180 (19.5) | | 111 (13.6) | | 855 (10.7) |
| **Stage; n (%)** | | <0.001 | | <0.001 | |
| I | 52 (5.6) | | 64 (7.9) | | 2448 (30.6) |
| II | 70 (7.6) | | 101 (12.4) | | 1467 (18.4) |
| III | 124 (13.4) | | 119 (14.6) | | 1022 (12.8) |
| IV | 587 (63.5) | | 477 (58.7) | | 2647 (33.1) |
| Missing | 92 (10.0) | | 52 (6.4) | | 411 (5.1) |

*p-value for dead within 6 months as compared to those alive after 12 months

**Table 4. Univariate analysis, targeted variable is death within 6 months.**

| Variable | Dead within 6 months/N | Dead within 6 months % (95% KI) | RR (95% KI) | P* |
|---|---|---|---|---|
| **Diagnosis** | | | | |
| Lip | 22/863 | 2.5 (1.6–3.8) | 0.23 (0.15–0.36) | <0.001 |
| Oral Cavity | 298/2714 | 10.9 (9.8–12.2) | 1.0 | - |
| Oropharynx | 217/2524 | 8.6 (7.5–9.8) | 0.78 (0.66–0.92) | 0.004 |
| Nasopharynx | 10/234 | 4.3 (2.1–7.7) | 0.39 (0.21–0.72) | 0.003 |
| Hypopharynx | 106/455 | 23.3 (19.5–27.5) | 2.12 (1.74–2.59) | <0.001 |
| Larynx | 105/1300 | 8.1 (6.6–9.7) | 0.74 (0.59–0.91) | 0.005 |
| Nose/Sinuses | 66/477 | 13.8 (10.9–17.3) | 1.26 (0.98–1.61) | 0.068 |
| Salivary Gland | 51/795 | 6.4 (4.8–8.3) | 0.58 (0.44–0.78) | <0.001 |
| CUP | 46/355 | 13.0 (9.6–16.9) | 1.18 (0.88–1.58) | 0.263 |
| **Sex** | | | | |
| Men | 589/6244 | 9.4 (8.7–10.2) | 1.0 | - |
| Women | 336/3489 | 9.6 (8.7–10.7) | 1.02 (0.90–1.16) | 0.750 |
| **Age (continuously)** | 925/9733 | 9.5 (8.9–10.1) | 1.06 (1.05–1.06) | <0.001 |
| **Age groups** | | | | |
| 0–39 | 0/295 | 0.0 (0.0–1.2) | - | - |
| 40–49 | 10/644 | 1.6 (0.7–2.8) | 0.44 (0.23–0.85) | 0.015 |
| 50–59 | 62/1751 | 3.5 (2.7–4.5) | 1.0 | - |
| 60–69 | 226/3165 | 7.1 (6.3–8.1) | 2.02 (1.54–2.67) | <0.001 |
| 70–79 | 266/2217 | 12.0 (10.7–13.4) | 3.39 (2.59–4.44) | <0.001 |
| 80- | 361/1661 | 21.7 (19.8–23.8) | 6.14 (4.73–7.97) | <0.001 |
| **Stage** | | | | |
| I | 52/2564 | 2.0 (1.5–2.7) | 1.0 | - |
| II | 79/1638 | 4.3 (3.3–5.4) | 2.11 (1.48–3.00) | <0.001 |
| III | 124/1265 | 9.8 (8.2–11.6) | 4.83 (3.52–6.63) | <0.001 |
| IV | 587/3711 | 15.8 (14.7–17.0) | 7.80 (5.90–30.3) | <0.001 |
| **WHO score** | | | | |
| 0 | 147/6223 | 2.4 (2.0–2.8) | 1.0 | - |
| 1 | 166/1279 | 13.0 (11.1–14.9) | 5.49 (4.44–6.80) | <0.001 |
| 2 | 158/608 | 26.0 (22.5–29.7) | 11.0 (8.93–13.6) | <0.001 |
| 3 | 167/344 | 48.5 (43.2–54.0) | 20.6 (16.9–24.9) | <0.001 |
| 4 | 107/133 | 80.5 (72.7–86.8) | 34.1 (28.4–40.8) | <0.001 |
| Unknown | 180/1146 | 15.7 (13.6–18.0) | 6.65 (5.40–8.19) | <0.001 |
| **Treatment intention** | | | | |
| Curative | 396/8751 | 4.5 (4.1–5.0) | 1.0 | - |
| Palliative | 478/882 | 54.2 (50.8–57.5) | 12.0 (10.7–13.4) | <0.001 |
| **Given treatment** | | | | |
| Surgery only | 82/2370 | 3.5 (2.8–4.3) | 1.0 | - |
| Surgery+postoperative RT** | 24/1295 | 1.8 (1.2–27.4) | 0.54 (0.34–0.84) | 0.007 |
| Preoperative RT+surgery | 3/306 | 1.0 (0.2–2.8) | 0.28 (0.09–0.89) | 0.031 |
| Surgery+RT | 5/356 | 1.4 (0.5–3.3) | 0.41 (0.17–0.99) | 0.049 |
| RT only | 300/2625 | 11.4 (10.2–12.7) | 3.30 (2.60–4.19) | <0.001 |
| RT+Chemotherapy | 43/1169 | 3.7 (2.7–4.9) | 1.06 (0.74–1.53) | 0.741 |
| Other treatment | 38/818 | 4.8 (3.4–6.5) | 1.34 (0.92–1.96) | 0.125 |
| No treatment | 169/245 | 52.4 (48.9–55.9) | 13.7 (10.9–17.3) | <0.001 |

*P-value is for Relative Risk

**RT = radiotherapy

**Table 5. Multivariable analysis, targeted variable is death within 6 months (n = 9098).**

| Variable | Curative and palliative treatment (n = 9098) | | Curative treatment (n = 8314) | |
|---|---|---|---|---|
| | RR (95% CI) | P* | RR (95% CI) | P* |
| **Diagnosis** | | | | |
| Lip | 0.76 (0.49–1.19) | 0.227 | 0.69 (0.41–1.17) | 0.167 |
| Oral Cavity | 1.0 | - | 1.0 | - |
| Oropharynx | 0.96 (0.82–1.13) | 0.645 | 0.91 (0.68–1.20) | 0.500 |
| Nasopharynx | 0.61 (0.33–1.12) | 0.109 | 0.38 (0.09–1.52) | 0.171 |
| Hypopharynx | 1.24 (1.03–1.50) | 0.024 | 1.27 (0.90–1.78) | 0.179 |
| Larynx | 1.09 (0.89–1.33) | 0.424 | 1.09 (0.81–1.47) | 0.583 |
| Nose/Sinuses | 0.94 (0.75–1.19) | 0.624 | 1.02 (0.66–1.58) | 0.930 |
| Salivary Gland | 0.59 (0.45–0.79) | <0.001 | 0.63 (0.41–0.99) | 0.046 |
| **Sex** | | | | |
| Men | 1.0 | - | 1.0 | - |
| Women | 0.96 (0.85–1.09) | 0.518 | 0.81 (0.65–1.01) | 0.060 |
| **Age (continuous)** | 1.023 (1.017–1.029) | <0.001 | 1.039 (1.029–1.049) | <0.001 |
| **Stage** | | | | |
| I | 1.0 | - | 1.0 | - |
| II | 1.68 (1.17–2.40) | 0.005 | 1.59 (1.03–2.48) | 0.030 |
| III | 2.82 (2.02–3.93) | <0.001 | 3.22 (2.14–4.85) | <0.001 |
| IV | 3.71 (2.71–5.10) | <0.001 | 4.61 (3.17–6.70) | <0.001 |
| **WHO score** | | | | |
| 0 | 1.0 | - | 1.0 | - |
| 1 | 3.37 (2.67–4.25) | <0.001 | 3.25 (2.45–4.31) | <0.001 |
| 2 | 4.53 (3.54–5.80) | <0.001 | 4.82 (3.51–6.62) | <0.001 |
| 3 | 5.68 (4.40–7.32) | <0.001 | 8.29 (5.92–11.6) | <0.001 |
| 4 | 6.77 (5.22–8.78) | <0.001 | 16.8 (10.4–21.4) | <0.001 |
| unknown | 3.47 (2.73–4.40) | <0.001 | 3.08 (2.25–4.23) | <0.001 |
| **Treatment intention** | | | | |
| Curative | 1.0 | - | | |
| Palliative | 3.16 (2.68–3.72) | <0.001 | | |

*P-value is for relative risk

4, 107 died within 6 months (80.5%). The relative risk of death was 3.37 times higher for patients with a WHO score of 1 (237%) compared with those who scored 0, Table 6.

## Tumour-related factors

**Site:** the probability of survival varied depending on the site of the primary tumour (Fig 3). The worst prognosis was found among patients with hypopharyngeal cancer, of whom 106 died within six months (23%), Table 4. Patients with hypopharyngeal cancers had an increased relative risk for early death of 1.24 compared with those with cancer of the oral cavity (p = 0.024), Table 6.

Salivary gland cancer is morphologically considered to be a separate entity of head and neck cancer. Therefore, we performed analyses excluding salivary gland cancer in order to evaluate differences in risk of death within six months. The exclusion of salivary gland cancer did not significantly affect the outcome, Tables 6 and 7.

**Table 6. Multivariate analysis, targeted variable is death within 6 months, including all tumour sites (n = 9098).**

| Variable | RR (95% CI) | P* |
|---|---|---|
| **Diagnosis** | | |
| Lip | 0.76 (0.49–1.19) | 0.227 |
| Oral Cavity | 1.0 | - |
| Oropharynx | 0.96 (0.82–1.13) | 0.645 |
| Nasopharynx | 0.61 (0.33–1.12) | 0.109 |
| Hypopharynx | 1.24 (1.03–1.50) | 0.024 |
| Larynx | 1.09 (0.89–1.33) | 0.424 |
| Nose/Sinuses | 0.94 (0.75–1.19) | 0.624 |
| Salivary Gland | 0.59 (0.45–0.79) | <0.001 |
| **Sex** | | |
| Men | 1.0 | - |
| Women | 0.96 (0.85–1.09) | 0.518 |
| **Age (continuous)** | 1.023 (1.017–1.029) | <0.001 |
| **Stage** | | |
| I | 1.0 | - |
| II | 1.68 (1.17–2.40) | 0.005 |
| III | 2.81 (2.01–3.92) | <0.001 |
| IV | 3.74 (2.71–5.13) | <0.001 |
| **WHO score** | | |
| 0 | 1.0 | - |
| 1 | 3.37 (2.67–4.25) | <0.001 |
| 2 | 4.53 (3.54–5.80) | <0.001 |
| 3 | 5.68 (4.40–7.32) | <0.001 |
| 4 | 6.77 (5.22–8.78) | <0.001 |
| unknown | 3.47 (2.73–4.40) | <0.001 |
| **Treatment intention** | | |
| Curative | 1.0 | - |
| Palliative | 3.16 (2.68–3.72) | <0.001 |

*P-value for relative risk of death within 6 months

**Stage:** we found an association between tumour stage and the risk of death within six months, Table 4. Among the 925 patients who died within six months, 587 had stage IV disease (64%), and only 52 patients (6%) who died within six months had stage I tumours. The distribution between tumour stages also varied depending on the site of the primary tumour with for example, 699 of all cancers of the lip (81%) being stage I, and only 17 (2%) stage IV. Stage IV disease was found at diagnosis in two-thirds of patients with oropharyngeal (n = 1691, 67%) and hypopharyngeal (n = 305, 66%) cancers. Most of the 152 patients with nasopharyngeal cancer (65%) were diagnosed with stage III or IV disease. In total, a patient with stage IV disease had a 3.7 times higher relative risk of death within six months than patients with stage I disease (p<0.001), Table 6.

**TNM class.**

T: low T class correlated with a better prognosis. Only 77 patients with a T1 tumour (2.4%) died within six months, while the six-month mortality among T4 patients was 455 (23%).

N: 5777 of patients (59%) had no neck metastases, whereas 3791 did (39%). Data about N class were missing in 2% of patients. A total of 399 patients with N negative necks (7%) died within six months, as did 482 patients (13%) with N positive necks.

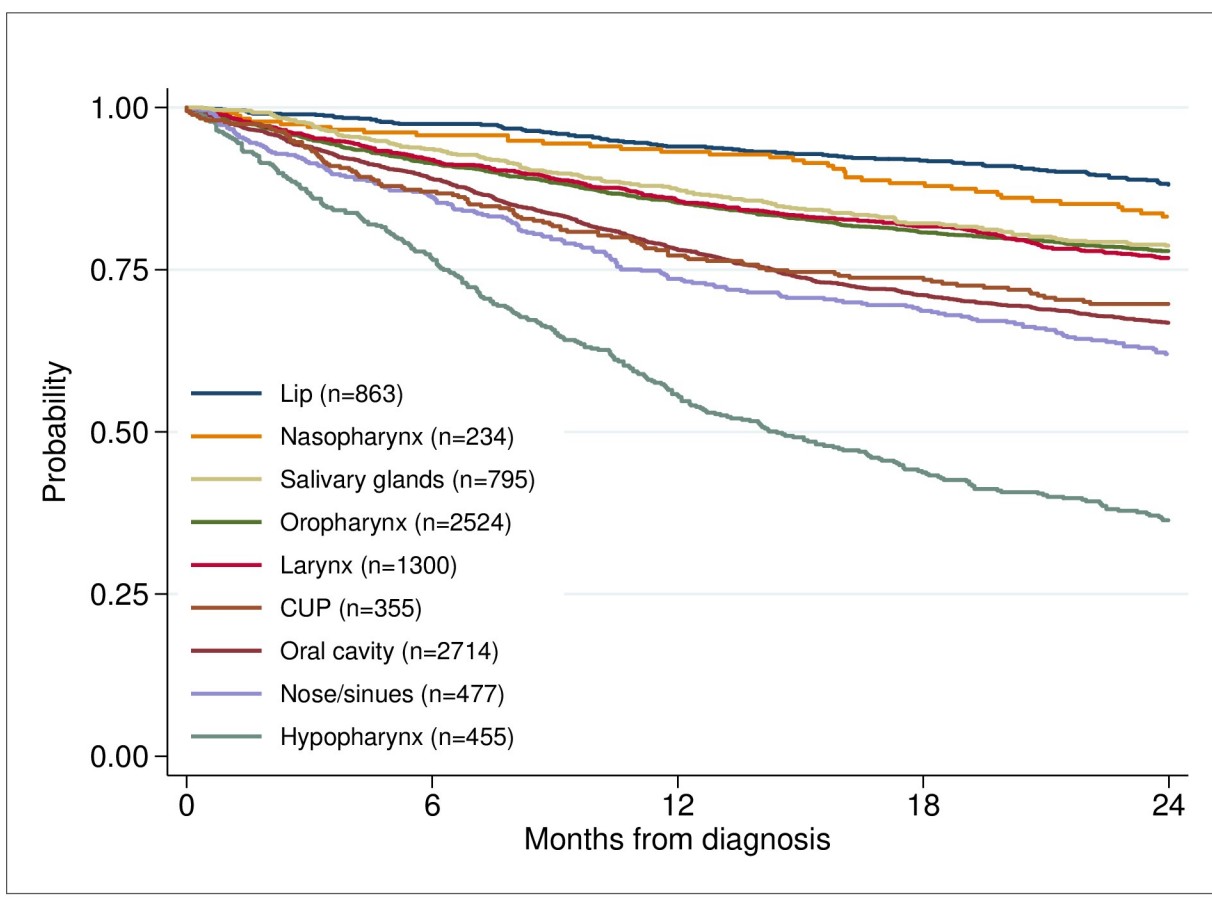

**Fig 3. Probability of survival based on site of tumor.**

<u>M</u>: Of patients without distant metastases 753 (8%) died within six months. A total of 256 patients (3%) had distant metastasis at the time of diagnosis, and of these 125 (49%) died within six months.

## Treatment-related factors

**Curative compared with palliative intent.** Treatment with curative intent was recommended to 8751 patients (90.8%), and palliative treatment to 882 (9.2%) (Fig 2). Among those whose treatment was potentially curative, 396 (4.5%) died within six months (Fig 1, Table 4). Risk of early death increased among patients with curative treatment decision and high WHO score and/or high stage, Table 5. Of those for whom palliative treatment or "best supportive care" was recommended, 478 died within six months (54.2%). During the period 2008–2014 where 8271 patients were studied and a two-year follow up was available interestingly 104 patients (14%) for whom palliative treatment had been recommended survived for more than two years.

**Treatment: Modality.** A total of 2370 patients (24%) were treated by primary surgery alone, and primary surgery was combined with postoperative radiotherapy (RT) in 1295 (16%) (Fig 4).

In the group who had primary surgery alone, 82 (3.5%) died within six months. When it was combined with postoperative RT, 24 (1.8%) died within six months. Preoperative RT or chemoradiotherapy (CRT) together with surgery resulted in the death of eight patients (1%)

**Table 7. Multivariate analysis, targeted variable is death within 6 months, excluding salivary gland cancer (n = 8332).**

| Variable | RR (95% CI) | P* |
|---|---|---|
| **Diagnosis** | | |
| Lip | 0.77 (0.49–1.21) | 0.255 |
| Oral Cavity | 1.0 | - |
| Oropharynx | 0.97 (0.83–1.13) | 0.663 |
| Nasopharynx | 0.60 (0.33–1.11) | 0.103 |
| Hypopharynx | 1.24 (1.03–1.50) | 0.023 |
| Larynx | 1.09 (0.89–1.34) | 0.383 |
| Nose/Sinuses | 0.94 (0.75–1.19) | 0.619 |
| **Sex** | | |
| Men | 1.0 | - |
| Women | 0.99 (0.87–1.12) | 0.863 |
| **Age (continuous)** | 1.022 (1.016–1.028) | <0.001 |
| **Stage** | | |
| I | 1.0 | - |
| II | 1.75 (1.21–2.52) | 0.003 |
| III | 2.88 (2.05–4.07) | <0.001 |
| IV | 3.72 (2.68–5.16) | <0.001 |
| **WHO score** | | |
| 0 | 1.0 | - |
| 1 | 3.40 (2.68–4.31) | <0.001 |
| 2 | 4.55 (3.54–5.86) | <0.001 |
| 3 | 5.59 (4.30–7.26) | <0.001 |
| 4 | 6.48 (4.96–8.45) | <0.001 |
| unknown | 3.41 (2.66–4.37) | <0.001 |
| **Treatment intention** | | |
| Curative | 1.0 | - |
| Palliative | 3.28 (2.77–3.88) | <0.001 |

*P-value is for relative risk

within six months. Patients given either RT or CRT alone had a mortality of 343 within six months (9%), Table 4.

**Treatment: Outcome.** Patients in whom locoregional control was achieved after primary treatment had a low risk of early death, and only 52 died within six months (0.7%). Among those for whom primary treatment failed, 381 (35%) died within six months. For a total of 1236 patients (12.7%), data about locoregional control after treatment were missing, and for these patients the mortality was 493 within six months (40%).

**Patient–tumour, and treatment-related factors.** Taken together, patient-, tumour and treatment- related factors all influenced the risk of death within six months. In order to clarify the extent of how vastly it affects this risk, a comparison between a low-risk patient (55-year old, oral cavity cancer, WHO 0, stage 1, curative treatment) and a high-risk patient (75-year old, hypohryngeal cancer, WHO 2, stage III, curative treatment) was made and showed a 24.9 fold risk increase to die within six months for the high-risk patient (95% CI: 16.3–38.0, p<0.001). If the low-risk patient however, had a stage III disease instead of stage I, the risk of early death was still 8.8 times higher for the 75-year old man (95% CI: 6.47–12.06, p<0.001).

**Recurrence.** A total of 8437 of the patients (87%) had no recurrence within the first 12 months.

## Discussion

Studies on the incidence and aetiology of early death among patients with cancers of the head and neck are still scarce, and to our knowledge this is not only the first official report to describe the SweHNCR data on 9733 such patients, but also the first attempt in Europe to investigate early death among patients in a nationwide population-based series. A previous Swedish study based on the data from 6785 such patients during the period 2008–2013 indicated a risk of death within six months of 665 (9.8%) [9]. In the present study we found a slightly reduced risk of early death of 925/9733 patients during the extended period 2008–2015 (9.5%). If data from the group who were given palliative treatment were excluded, the remainder (those given curative treatment) had a six-month mortality of 396 (4.5%). Our results suggest that extra precautions should be taken for patients with tumours located in the hypopharyngeal area, stage IV disease, older age, and increased WHO score.

Five-year mortality from cancers of the head and neck is high, even though mortality in general has decreased in Northern Europe. One reason for this is the increase in the number of HPV-induced tumours [23, 24]. Patients in the SweHNCR have an overall relative five-year survival rate of 6521 (67%), which includes all sites [16]. It has been reported that patients with early stage cancers of the head and neck can have five-year survival rates of up to 80%, whereas those with late-stage disease have reduced five-year survival of about 20% [25]. Early mortality (within six months of diagnosis) has been studied in other cancers [26, 27] but to our knowledge still remains unexplored for the head and neck.

Our results indicate that the risk of early death was associated with patient- related and treatment-related factors as well as tumour-related ones. Eskiizmir et al in 2017 showed differences in survival by sex in early-stage laryngeal cancer [21], but we could not confirm a sex difference in stage I-II laryngeal cancer in our series as 336 (9.6%) women and 590 (9.4%) men died within six months (p = 0.518).

Our results showed that older patients with cancers of the head and neck had a poorer prognosis than younger ones, and that the risk of early death increased with older age. Other authors have also found that mortality increases for older patients after resection and CRT [22, 23]. A Danish study by Johansen et al, found that survival improved from 1980 to 2012 for all age groups up to 79 years of age, but this effect was less pronounced for patients over 80 [24]. Prediction of survival among older patients must be based on available demographic information, and the details that are recorded differ among countries. We found a total of 1661 patients over 80 years old, of whom 361 (22%) died within six months. Multivariate analysis showed that patients over 80 were more likely to have coexisting conditions, but up to 75 years of age most patients who had a WHO score of 0–1 were less likely to die early. Older patients with cancer given curative treatment, therefore, should be particularly carefully monitored to minimise complications and failure of treatment, and ultimately, the risk of early death.

A high WHO score has been reported to be an indicator of a poor prognosis in both leukaemia and colon cancer [25, 26]. A total of 107 patients with cancer of the head and neck and a WHO score of 4 (80.5%) died within six months, Table 4. Among patients who had a WHO score of 4, 109 (82%) were given palliative treatment because of the severity of their disease, or coexisting conditions, or both.

Cancers of the head and neck are a heterogeneous group of malignancies with varied biology. The aggressiveness of the tumour, coexisting conditions, and presenting symptoms can all affect the outcome. For example, patients who presented with easily-detected symptoms such as those of lip cancer (801 of 836 tumours (93%) were diagnosed in stages I-II) or laryngeal cancer (819 of 1280 tumours, (63%) were diagnosed in stages I-II), had better survival than those with cancers at sites associated with more subtle symptoms, such as the

hypopharynx. Patients with hypopharyngeal tumours tended to be detected at later stages (293 of 341 tumours (86%) were diagnosed at stages III-IV). If the tumours were discovered later, patients were at higher risk of early death, findings confirmed elsewhere [28]. Stage IV cancer had a 3.7 fold risk of death within six months, compared with those with stage I (p<0.001).

Patients with advanced tumours had the shortest survival, independent of the site of the tumour. In accordance with a number of previous publications [29, 30], our results showed that advanced tumours and metastatic disease were both associated with an increased risk of early death. However, there was a significant difference in the risk of early death depending on the site of the tumour–for example, a patient with hypopharyngeal cancer had a 24% increased risk compared to patients with tumours in the oral cavity (p<0.024). This is in line with the results from more than 34 000 patients from Denmark in 2014, among whom the probability of five-year survival for those with hypopharyngeal cancer was only 22.8% compared with 62.4% for all other sites in the head and neck [31].

Patients being given treatment with curative intent lived longer than those who were given palliation, who were at a 12-fold risk of death within six months (p<0.001). However, it should be noted that 223 of patients who had palliative treatment (25%) lived for longer than 12 months, and 104 (14%) for 24 months or more, which is longer than could be expected. This is particularly interesting as other authors have found that without any treatment the median survival was usually less than four months [3]. These data suggest that even if the intention of treatment is not to cure but to palliate, palliative treatment can be effective for a substantial subgroup, and even prolong time to death.

Patients who were operated on primarily had smaller tumours (1975 of them (87%) had stage I or II disease). They also had tumours in earlier stages than patients who were treated by radiotherapy or CRT together with resection, and were therefore more likely to survive. It has been reported that higher doses of chemotherapy and radiotherapy together with extensive resection will increase the risk of lethal complications [32]. Patients with advanced cancers of the head and neck who were admitted to hospital for more than five days during or after radiotherapy had worse locoregional control, progression-free survival, and overall survival [33].

In the group of patients (n = 245) who were given no active treatment, 89 (36%) had been recommended for curative treatment at a multidisciplinary tumour board meeting, and 43 (48%) of them died within six months. It could be speculated that these patients chose not to undergo treatment, were too ill to start, or died before treatment could begin.

For the 1236 patients (12.7%) with missing data about locoregional control at first follow-up, the mortality within six months was high 493 (40%). At the first check-up after treatment it can be difficult to decide whether a patient has a residual tumour, or just divergent anatomy/mucosa after intense treatment. As mortality is so high in this group, clinicians could be more cautious, and suspect recurrence as soon as there is any doubt.

A limitation of this study was that the actual cause of death is not recorded in the SweHNCR. It is recorded if the patient is considered to be free of tumour, but the immediate cause of death is not reported. Questions about whether death was caused by the cancer itself, by treatment-related complications, coexisting conditions, or causes unrelated to the cancer, could therefore not be answered by this study. Other limiting factors include the lack of information about socioeconomic factors, alcohol consumption, smoking habits, and HPV status, which were not reported in the SweHNCR until 2015, but thereafter alcohol consumption, smoking habits, and HPV status have been included. Sawabe et al showed that a high alcohol consumption leads to significantly shorter overall survival for these patients [34]. Socioeconomic status, other life-style factors, and depression have also been suggested as risk factors for early death [35–39]. As these data were not reported in the SweHNCR during the study period, we were unable to evaluate them.

Smoking is decreasing in Sweden, and roughly 9% of the total population now smoke daily compared with 15% 10 years ago (according to the Public Health Agency in Sweden). One could speculate that over time this would result in a decrease in mortality as well. The incidence of HPV-related cancer in the oropharynx is increasing, which can influence and decrease mortality [40] [41]. However, the large group of patients, its homogeneity in gross socioeconomic and health-related factors from a well-defined geographical area, and the degree of coverage of 98.5%, compensate for some of the limiting factors.

We found that older age, advanced stage, and a hypopharyngeal tumour, together with higher WHO score and palliative treatment were significant risk factors for early death. These findings contribute to the knowledge of a risk profile for early death among patients with cancers of the head and neck, and we think it is important to identify these patients early to optimise treatment, control symptoms, and reduce the number who die early. This study could be used as a base for future studies of subgroups within the palliative and curative treatment groups, so that we are able to understand better which patients are at greatest risk of early death.

Taken together, this study indicates that from clinical relevant information, risk profiles can be identified, and a high-risk patient had a considerable increased risk to die within six months compared to a low-risk patient. Understanding and identifying high-risk patients for early death is therefore important for physicians when recommending treatment to patients, curative or palliative.

## Conclusion

We identified a risk profile for early death in this population-based register study of 9733 patients with cancers of the head and neck. The risk depended on patient's age, WHO score, site of primary tumour, stage, and intention to treat. Our most important findings were that the risk of early death increased with 2.3% for every year older the patient became (p<0.001). The relative risk of death was 3.37 times higher for patients with a WHO score of 1, compared with a WHO score of 0 (p<0.001). Patients with stage IV disease had a 3.7 times greater risk of death within six months than those with stage I disease (p<0.001), and the group given palliative treatment had a 12 times higher risk of early death (p<0.001) than the others. If the tumour was located in the hypopharynx the risk of dying within six months increased by 24% compared with that of a tumour in the oral cavity (p = 0.024). Even if the intention of treatment was not cure but palliation, palliative treatment was shown to have an effect for a substantial group, and possibly prolong time to death.

## Acknowledgments

We thank the Swedish regional cancer centers and the board of SweHNCR for supporting this study and providing register data.

## Author Contributions

**Conceptualization:** Charbél Talani, Antti Mäkitie, Göran Laurell, Lovisa Farnebo.

**Data curation:** Göran Laurell, Lovisa Farnebo.

**Formal analysis:** Charbél Talani, Antti Mäkitie, Lovisa Farnebo.

**Investigation:** Antti Mäkitie, Lovisa Farnebo.

**Methodology:** Charbél Talani, Antti Mäkitie, Erik Holmberg, Göran Laurell, Lovisa Farnebo.

**Project administration:** Charbél Talani, Antti Mäkitie, Göran Laurell, Lovisa Farnebo.

**Resources:** Lovisa Farnebo.

**Software:** Erik Holmberg.

**Supervision:** Antti Mäkitie, Martin Beran, Erik Holmberg, Göran Laurell, Lovisa Farnebo.

**Validation:** Charbél Talani, Martin Beran, Erik Holmberg, Göran Laurell, Lovisa Farnebo.

**Visualization:** Göran Laurell, Lovisa Farnebo.

**Writing – original draft:** Antti Mäkitie, Göran Laurell, Lovisa Farnebo.

**Writing – review & editing:** Charbél Talani, Antti Mäkitie, Martin Beran, Erik Holmberg, Göran Laurell, Lovisa Farnebo.

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
