## [Decision Letter · Decision Letter 0]

10 Jul 2019

PONE-D-19-15950

Early mortality after diagnosis of cancer of the head and neck – A population-based nationwide study

PLOS ONE

Dear MD, PhD Farnebo,

Thank you for submitting your manuscript to PLOS ONE. After careful consideration, we feel that it has merit but does not fully meet PLOS ONE’s publication criteria as it currently stands. Therefore, we invite you to submit a revised version of the manuscript that addresses the points raised during the review process.

For acceptance the revision will need to include modification of the methods section and improved statistical review as documented by reviewer one.  I also agree with reviewer 2 that salivary gland cancers can have entirely different histologies and therefore prognosis from SCC and therefore would consider excluding them or doing a sub-analysis excluding them that is reported in the manuscript.  Finally, reviewer 2 mentions that it may be helpful to do a sub-analysis of those receiving curative treatment to see who does not benefit - while not mandatory for publication, I agree could be an interesting finding.

We would appreciate receiving your revised manuscript by Aug 24 2019 11:59PM. To enhance the reproducibility of your results, we recommend that if applicable you deposit your laboratory protocols in protocols.io, where a protocol can be assigned its own identifier (DOI) such that it can be cited independently in the future. For instructions see: http://journals.plos.org/plosone/s/submission-guidelines#loc-laboratory-protocols

We look forward to receiving your revised manuscript.

Kind regards,

Jessica D McDermott, MD, MSCS

Academic Editor

PLOS ONE

Journal Requirements:

2. Please include your tables as part of your main manuscript and remove the individual files. Please note that supplementary tables (should remain/ be uploaded) as separate "supporting information" files

3. Please ensure that you refer to Figure 4 in your text as, if accepted, production will need this reference to link the reader to the figure.

Reviewers' comments:

Reviewer's Responses to Questions

**Comments to the Author**

1. Is the manuscript technically sound, and do the data support the conclusions?

Reviewer #1: Partly

Reviewer #2: Yes

2. Has the statistical analysis been performed appropriately and rigorously? 

Reviewer #1: No

Reviewer #2: Yes

3. Have the authors made all data underlying the findings in their manuscript fully available?

Reviewer #1: No

Reviewer #2: Yes

4. Is the manuscript presented in an intelligible fashion and written in standard English?

Reviewer #1: Yes

Reviewer #2: Yes

5. Review Comments to the Author

Reviewer #1: Overall an interesting article which some gaps in the methodological presentation. Also, the presentation of the results could benefit from being more precise because otherwise they sound in part trivial, which is not really the case.

Methods:

The description of the statistical analysis is presented in four lines. Given that by using this method all results have been obtained this looks odd. More details need to be added and also relevant literature needs to be cited. So far only a citation to Stata is provided which is a software package. Please cite original papers introducing the used methods.

It seems no Cox proportional hazard model has been used. Why is it not appropriate in this case?

What statistical test has been used to identify a difference in the survival times? Why has this test been selected (discuss and compare with alternatives)?

A discussion about censoring is completely missing. How has this been handeled? What type of censoring is present?

Results:

The figures of the survival curves do not include information about censoring.

For all statistical test, please add the actual p-values and not just p<0.xyz.

Conclusions:

Identification of patients at i 52 ncreased risk of early death shows that older

53 patients with advanced disease, increased WHO score, primary tumour in the hypopharynx,

54 and those given palliative treatment, are more likely than the others to die from head and neck

55 cancer within six months of diagnosis.

This does not sound surprising. What aspect of this is none trivial? Adding quantitive results would make this actually interesting.

We think that it would be

327 beneficial to identify patients at risk early and bear this information in mind when planning

328 their treatment and follow up.

I wonder if there is any disease for which this would NOT be beneficial? If there is none, this is trivial.

Reviewer #2: The authors reported that old age, advanced stage, poor performance status and no curative treatment were the clinical risk factors associated with early death 6 months after diagnosis of head and neck cancer using a nationwide data although the incidence and case number of head and neck cancer are relatively smaller than those of other countries. It is really helpful for clinicians to think about who are not the candidate to recieve the curative treatment, especially in old people or people with poor performance so that will be better if this study describes or analysis the outcome of these people with curative treatment.

Salivary gland cancer is also a totally different cancer from other head and neck cancer so the story is possibly different if salivary gland cancer is excluded.

6. PLOS authors have the option to publish the peer review history of their article (what does this mean?). If published, this will include your full peer review and any attached files.

Reviewer #1: No

Reviewer #2: No

---

## [Author Response · Author response to Decision Letter 0]

14 Aug 2019

Responses to the Reviewers

Thank you for the received Reviewer reports and the opportunity to revise our manuscript. We have now improved the manuscript to include a more thorough Methods section and we also upgraded the description of the statistical review. Furthermore, we included a new sub-analysis excluding salivary gland cancers, and included a cohort with only curative treatment intent, as recommended by Reviewer 2. In order to meet the request on a sub-analysis of those receiving curative treatment to see who does not benefit, we also added examples of a low-risk patient and a patient with high risk of six-month mortality, in order to emphasize the conclusion in the manuscript.

Our detailed responses to the Reviewers are listed below in blue, and corrections of the manuscript appear highlighted in yellow in the revised version. 

2. Please include your tables as part of your main manuscript and remove the individual files. Please note that supplementary tables (should remain/ be uploaded) as separate "supporting information" files

Response: Tables are now included in the revised version of the main manuscript, and individual files removed.

3. Please ensure that you refer to Figure 4 in your text as, if accepted, production will need this reference to link the reader to the figure.

Response: This has now been done accordingly and we refer to Figure 4 in the Results section (page 10, paragraph 2, row 198).

Comments to the Author

1. Is the manuscript technically sound, and do the data support the conclusions?

Reviewer #1: Partly 

Reviewer #2: Yes

Response: We have improved the Methods section accordingly and we included new statistical calculations to improve the manuscript, page 7.

2. Has the statistical analysis been performed appropriately and rigorously? 

Reviewer #1: No 

Reviewer #2: Yes

Response: We have taken measures to improve the clarity in the Methods section. Statistical analyses have been described and explained in more detail and new calculations have been added, and are highlighted in yellow on pages 7, 9, 10, 11 and 16.

3. Have the authors made all data underlying the findings in their manuscript fully available?

Reviewer #1: No

Reviewer #2: Yes

Response: According to the approval of this study by the Swedish Ethical Review Authority we are not allowed to make data available in any form other than aggregated data. 

4. Is the manuscript presented in an intelligible fashion and written in standard English?

Reviewer #1: Yes

Reviewer #2: Yes

5. Review Comments to the Author

Reviewer #1: Overall an interesting article which some gaps in the methodological presentation. Also, the presentation of the results could benefit from being more precise because otherwise they sound in part trivial, which is not really the case.

Response: We have improved the methodological presentation in text and tables. In order to avoid the Results section from sounding trivial, we have included an example showing difference in RR between a low-risk patient and a high-risk patient, page 11. We hope this addition emphasizes the value of the conclusions in this article.

Methods:

The description of the statistical analysis is presented in four lines. Given that by using this method all results have been obtained this looks odd. More details need to be added and also relevant literature needs to be cited. So far only a citation to Stata is provided which is a software package. Please cite original papers introducing the used methods.

Response: The statistical analyses have been explained in more detail and relevant citations (see below) are now included in the revised version of our paper.

1. Schmidt CO, Kohlmann T. When to use the odds ratio or the relative risk? Int J Public Health 53 (2008) 165–167.

2. Kaplan EL, Meier P. Nonparametric estimation from incomplete observations. Journal of the American Statistical Association 53 (1958) 457–481.

3. Cuzick J. 1985. A Wilcoxon-type test for trend. Statistics in Medicine 4 (1985) 87-90.

4. Clopper CJ and Pearson ES. The use of confidence or fiducial limits illustrated in the case of binomial. Biometrika 26 (1934) 404-413.

5. StataCorp. 2017. Stata: Release 15. Statistical Software. College Station, TX: StataCorp LLC.

It seems no Cox proportional hazard model has been used. Why is it not appropriate in this case? What statistical test has been used to identify a difference in the survival times? Why has this test been selected (discuss and compare with alternatives)? A discussion about censoring is completely missing. How has this been handeled? What type of censoring is present?

Response: Cox proportional hazard model is mainly recommended when censoring needs handling. We have no censoring within the first 12 months of follow-up, which is why we instead used the Poisson-regression to analyze mortality within six months follow-up, a method suitable when no censoring needs handling.

The primary aim of this study was to analyze early mortality within 6 months. Since we had no censuring within 12 months follow-up the results were mainly presented as relative risk of death within 6 months follow-up. 

Differences in the 24 months survival have been presented as an illustrative example as Kaplan-Meier-curves in Figure 2-4 but have not been analyzed further. There are censuring after 12 months for cases due to end of follow-up which have been taken care of by the KM-method. There are no ‘lost to follow-up cases’ within 24 months. 

Results:

The figures of the survival curves do not include information about censoring.For all statistical test, please add the actual p-values and not just p<0.xyz.

Response: Information about censoring is to our knowledge, generally not presented in Kaplan-Meier diagrams. For clarity for the reader, we have not presented p-values of less than 0.001, we do however show 95%-confidence intervals, numbers that can give a deeper understanding than the p-value.

Conclusions:

Identification of patients at i 52 ncreased risk of early death shows that older

53 patients with advanced disease, increased WHO score, primary tumour in the hypopharynx,

54 and those given palliative treatment, are more likely than the others to die from head and neck

55 cancer within six months of diagnosis.

This does not sound surprising. What aspect of this is none trivial? Adding quantitive results would make this actually interesting.

Response: Comparison between a high-risk patient, a 75-year old with hypopharyngeal cancer WHO 2, stage III , curative treatment, and a low-risk patient, a 55-year old man with oral cavity cancer, WHO 0, stage I, curative treatment:

 ded_6 | IRR Std. Err. z P>|z| [95% Conf. Interval]

-------------+----------------------------------------------------------------

 (1) | 24.89902 5.375931 14.89 0.000 16.30797 38.01584

The relative risk RR of death within six months = 24.9 (95% CI: 16.3-38.0), p<0.001, for the high-risk patient compared to the low-risk.

If the 55-year old man instead had a stage III disease the RR = 8.83 (95% CI: 6.47 – 12.06), p<0.001

 ded_6 | IRR Std. Err. z P>|z| [95% Conf. Interval]

-------------+----------------------------------------------------------------

 (1) | 8.835795 1.402317 13.73 0.000 6.4737 12.05976

This means that a 75-year old hypopharyngeal cancer patient with WHO 2, stage III, had a 24.9 fold risk increase to die within six months as compared to a 55-year old with oral cavity cancer, WHO 0, stage 1. If the 55-year old however had a stage III disease, the risk of early death was 8.8 times higher for the 75-year old patient.

We think that it would be

327 beneficial to identify patients at risk early and bear this information in mind when planning

328 their treatment and follow up.

I wonder if there is any disease for which this would NOT be beneficial? If there is none, this is trivial.

We agree, this sentence is trivial, and it has now been removed from the manuscript and replaced with the high-risk/low-risk patient example above.

Reviewer #2: The authors reported that old age, advanced stage, poor performance status and no curative treatment were the clinical risk factors associated with early death 6 months after diagnosis of head and neck cancer using a nationwide data although the incidence and case number of head and neck cancer are relatively smaller than those of other countries. It is really helpful for clinicians to think about who are not the candidate to recieve the curative treatment, especially in old people or people with poor performance so that will be better if this study describes or analysis the outcome of these people with curative treatment.

Response: Table 5 has now been extended to include this information in Table 5a. Please find the new Table 5a, b, and c below. It can now be clearly visualized that higher WHO score, age, and stage all contribute to increased risk of death within 6 months in the patient group with curative treatment intent, information that clinicians should bear in mind at the multi-disciplinary tumour board meeting. 

Table 5a. Multivariable analysis, targeted variable is death within 6 months (n=9098)

 Curative and palliative treatment

(n=9098) Curative treatment

(n=8314)

Variable RR (95% CI) P* 

 RR (95% CI) P* 

Diagnosis

 Lip

 Oral Cavity

 Oropharynx

 Nasopharynx

 Hypopharynx

 Larynx

 Nose/Sinuses

 Salivary gland 

0.76 (0.49-1.19)

1.0

0.96 (0.82-1.13)

0.61 (0.33-1.12)

1.24 (1.03-1.50)

1.09 (0.89-1.33)

0.94 (0.75-1.19)

0.59 (0.45-0.79) 

0.227

-

0.645

0.109

0.024

0.424

0.624

<0.001 

0.69 (0.41-1.17)

1.0

0.91 (0.68-1.20)

0.38 (0.09-1.52)

1.27 (0.90-1.78)

1.09 (0.81-1.47)

1.02 (0.66-1.58)

0.63 (0.41-0.99) 

0.167

-

0.500

0.171

0.179

0.583

0.930

0.046

Sex

 Male

 Female 

1.0

0.96 (0.85-1.09) 

-

0.518 

1.0

0.81 (0.65-1.01) 

-

0.060

Age (continuous) 1.023 (1.017-1.029) <0.001 1.039 (1.029-1.049) <0.001

Stage

 I

 II

 III

 IV 

1.0

1.68 (1.17-2.40)

2.82 (2.02-3.93)

3.71 (2.71-5.10) 

-

0.005

<0.001

<0.001 

1.0

1.59 (1.03-2.48)

3.22 (2.14-4.85)

4.61 (3.17-6.70) 

-

0.030

<0.001

<0.001

WHO score

 0

 1

 2

 3

 4

 unknown 

1.0

3.37 (2.67-4.25)

4.53 (3.54-5.80)

5.68 (4.40-7.32)

6.77 (5.22-8.78)

3.47 (2.73-4.40) 

-

<0.001

<0.001

<0.001

<0.001

<0.001 

1.0

3.25 (2.45-4.31)

4.82 (3.51-6.62)

8.29 (5.92-11.6)

16.8 (10.4-21.4)

3.08 (2.25-4.23) 

-

<0.001

<0.001

<0.001

<0.001

<0.001

Treatment intent

 Curative

 Palliative 

1.0

3.16 (2.68-3.72) 

-

<0.001 

*P-value is for relative risk

Salivary gland cancer is also a totally different entity from other head and neck cancers so the story is possibly different if salivary gland cancer is excluded.

Response: We have now made an alternative analysis and excluded salivary gland cancer from the study cohort. It can be noted that no drastic statistical changes were seen when the patients with salivary gland cancer were excluded (see below Fig 5b, and 5c). 

 

Table 5b. Multivariate analysis, targeted variable is

death within 6 months (n=9098)

Variable RR (95% KI) P* 

Diagnosis

 Lip

 Oral Cavity

 Oropharynx

 Nasopharynx

 Hypopharynx

 Larynx

 Nose/Sinuses

 Salivary gland 

0.76 (0.49-1.19))

1.0

0.96 (0.82-1.13)

0.61 (0.33-1.13)

1.24 (1.03-1.50)

1.09 (0.89-1.33)

0.94 (0.75-1.19)

0.59 (0.45-0.79) 

0.227

-

0.645

0.109

0.024

0.424

0.624

<0.001

Sex

 Male

 Female 

1.0

0.96 (0.85-1.09) 

-

0.518

Age (continuous) 1.023 (1.017-1.029) <0.001

Stage

 I

 II

 III

 IV 

1.0

1.68 (1.17-2.40)

2.81 (2.01-3.92)

3.74 (2.71-5.13) 

-

0.005

<0.001

<0.001

WHO score

 0

 1

 2

 3

 4

 missing 

1.0

3.37 (2.67-4.25)

4.53 (3.54-5.80)

5.68 (4.40-7.32)

6.77 (5.25-8.83)

3.47 (2.73-4.40) 

-

<0.001

<0.001

<0.001

<0.001

<0.001

Treatment intent

 Curative

 Palliative 

1.0

3.16(2.68-3.72) 

-

<0.001

*P-value for relative risk of death within 6 months

Table 5c. Multivariate analysis, targeted variable is death within 6 months, 

excluding salivary gland cancer patients (n=8332)

Variable RR (95% KI) P* 

Diagnosis

 Lip

 Oral Cavity

 Oropharynx

 Nasopharynx

 Hypopharynx

 Larynx

 Nose/Sinuses 

0.77 (0.49-1.21)

1.0

0.97 (0.83-1.13)

0.60 (0.33-1.11)

1.24 (1.03-1.50)

1.09 (0.89-1.34)

0.94 (0.75-1.19) 

0.255

-

0.663

0.103

0.023

0.383

0.619

Sex

 Male

 Female 

1.0

0.99 (0.87-1.12) 

-

0.863

Age (continuous) 1.022 (1.016-1.028) <0.001

Stage

 I

 II

 III

 IV 

1.0

1.75 (1.21-2.52)

2.88 (2.05-4.07)

3.72 (2.68-5.16) 

-

0.003

<0.001

<0.001

WHO score

 0

 1

 2

 3

 4

 missing 

1.0

3.40 (2.68-4.31)

4.55 (3.54-5.86)

5.59 (4.30-7.26)

6.48 (4.96-8.45)

3.41 (2.66-4.37) 

-

<0.001

<0.001

<0.001

<0.001

<0.001

Treatment intent

 Curative

 Palliative 

1.0

3.28 (2.77-3.88) 

-

<0.001

*P-value is for relative risk

6. PLOS authors have the option to publish the peer review history of their article. If published, this will include your full peer review and any attached files.

Do you want your identity to be public for this peer review? For information about this choice, including consent withdrawal, please see our Privacy Policy.

Reviewer #1: No

Reviewer #2: No

---

## [Decision Letter · Decision Letter 1]

16 Sep 2019

Early mortality after diagnosis of cancer of the head and neck – A population-based nationwide study

PONE-D-19-15950R1

Dear Dr. Farnebo,

We are pleased to inform you that your manuscript has been judged scientifically suitable for publication and will be formally accepted for publication once it complies with all outstanding technical requirements.

With kind regards,

Jessica D McDermott, MD, MSCS

Academic Editor

PLOS ONE

Additional Editor Comments (optional):

Reviewers' comments:

Reviewer's Responses to Questions

**Comments to the Author**

1. If the authors have adequately addressed your comments raised in a previous round of review and you feel that this manuscript is now acceptable for publication, you may indicate that here to bypass the “Comments to the Author” section, enter your conflict of interest statement in the “Confidential to Editor” section, and submit your "Accept" recommendation.

Reviewer #2: All comments have been addressed

2. Is the manuscript technically sound, and do the data support the conclusions?

Reviewer #2: Yes

3. Has the statistical analysis been performed appropriately and rigorously? 

Reviewer #2: Yes

4. Have the authors made all data underlying the findings in their manuscript fully available?

Reviewer #2: Yes

5. Is the manuscript presented in an intelligible fashion and written in standard English?

Reviewer #2: Yes

6. Review Comments to the Author

Reviewer #2: I think that the authors answered all quesitons and the revised manuscript is well done and can be accepted for publication

7. PLOS authors have the option to publish the peer review history of their article (what does this mean?). If published, this will include your full peer review and any attached files.

Reviewer #2: No

---

## [Editor Report · Acceptance letter]

24 Sep 2019

PONE-D-19-15950R1 

Early mortality after diagnosis of cancer of the head and neck – A population-based nationwide study 

Dear Dr. Farnebo:

I am pleased to inform you that your manuscript has been deemed suitable for publication in PLOS ONE. Congratulations! Your manuscript is now with our production department. 

With kind regards,

on behalf of

Dr. Jessica D McDermott 

Academic Editor

PLOS ONE